# Genome-Wide Comparative Analysis of the *R2R3-MYB* Gene Family in Five Solanaceae Species and Identification of Members Regulating Carotenoid Biosynthesis in Wolfberry

**DOI:** 10.3390/ijms23042259

**Published:** 2022-02-18

**Authors:** Yue Yin, Cong Guo, Hongyan Shi, Jianhua Zhao, Fang Ma, Wei An, Xinru He, Qing Luo, Youlong Cao, Xiangqiang Zhan

**Affiliations:** 1State Key Laboratory of Crop Stress Biology for Arid Areas, College of Horticulture, Northwest A&F University, Yangling, Xianyang 712100, China; yinyue2011@nwafu.edu.cn (Y.Y.); guo_cong@nwafu.edu.cn (C.G.); shihongyan@nwafu.edu.cn (H.S.); housefang@nwafu.edu.cn (F.M.); 2National Wolfberry Engineering Research Center, Ningxia Academy of Agriculture and Forestry Sciences, Yinchuan 751002, China; zhaojianhua0943@163.com (J.Z.); 13037986722@163.com (W.A.); hhexinru@163.com (X.H.); luoqing640603@163.com (Q.L.)

**Keywords:** wolfberry, *R2R3-MYB* gene, carotenoids biosynthesis, expression analysis, co-expression

## Abstract

The *R2R3-MYB* is a large gene family involved in various plant functions, including carotenoid biosynthesis. However, this gene family lacks a comprehensive analysis in wolfberry (*Lycium barbarum* L.) and other Solanaceae species. The recent sequencing of the wolfberry genome provides an opportunity for investigating the organization and evolutionary characteristics of *R2R3-MYB* genes in wolfberry and other Solanaceae species. A total of 610 *R2R3-MYB* genes were identified in five Solanaceae species, including 137 in wolfberry. The *LbaR2R3-MYB* genes were grouped into 31 subgroups based on phylogenetic analysis, conserved gene structures, and motif composition. Five groups only of Solanaceae *R2R3-MYB* genes were functionally divergent during evolution. Dispersed and whole duplication events are critical for expanding the *R2R3-MYB* gene family. There were 287 orthologous gene pairs between wolfberry and the other four selected Solanaceae species. RNA-seq analysis identified the expression level of *LbaR2R3-MYB* differential gene expression (DEGs) and carotenoid biosynthesis genes (CBGs) in fruit development stages. The highly expressed *LbaR2R3-MYB* genes are co-expressed with CBGs during fruit development. A quantitative Real-Time (qRT)-PCR verified seven selected candidate genes. Thus, *Lba11g0183* and *Lba02g01219* are candidate genes regulating carotenoid biosynthesis in wolfberry. This study elucidates the evolution and function of *R2R3-MYB* genes in wolfberry and the four Solanaceae species.

## 1. Introduction

Wolfberry (*Lycium barbarum* L., 2n = 2x = 24), of the genus *Lycium* within the Solanaceae family, is an important Chinese traditional herbal medicine [1]. The fruits are a rich source of carotenoids, flavonoids, and polysaccharides, contributing to wolfberry’s immune-enhancing, antioxidant, and anti-tumor effects [2,3,4]. The carotenoids are responsible for *L. barbarum* fruit colorations [5,6]. Moreover, fruit color is a key factor in wolfberry fruit quality. Fruit colorations involve complex biochemical changes due to genetic and environmental factors. Hence, understanding the genetic factors controlling carotenoid accumulation is valuable to wolfberry breeding to generate novel fruit phenotypes. Carotenoid biosynthetic genes (CBGs) have been cloned and characterized in various plant species [7,8,9], including wolfberry. However, the transcriptional regulatory mechanisms of carotenoids in wolfberry are unclear.

V-myb avian myeloblastosis viral oncogene homolog (MYB) transcription factors (TFs) are major regulators of plant genes containing a highly conserved MYB DNA binding domain, widely distributed in eukaryotes [10]. The MYB superfamily is grouped into four subfamilies: 1R-MYB (MYB-related and R3-MYB), R2R3-MYB, 3R-MYB (R1R2R3-MYB), and 4RMYB. R2R3-MYB is the largest subfamily, with some members regulating plant growth and developmental [11], primary and secondary metabolism [12,13,14,15,16,17], response to various biotic and abiotic stresses [18,19,20,21], hormone synthesis, and signal transduction [22]. In recent years, more studies focused on the R2R3-MYB TFs regulating carotenoid metabolism. For example, the first R2R3-MYB TF, RCP1, regulates carotenoid pigmentation in *Mimulus lewisii* flowers [23]. In tomato, *SlMYB72* negatively regulates carotenoid biosynthesis by decreasing the expressions of phytoene synthase (*PSY)*, 15-cis-ζ-carotene isomerase (*ZISO)*, and lycopene β-cyclase (*LCYB)* genes [24]. In citrus, *CrMYB68* directly represses the transformation of α-and β-carotene by regulating the expressions of *CrBCH2* and *CrNCED5* promoters [25]. *AdMYB7* is overexpressed in kiwifruit, which regulates the *LCY-β* promoter, thus increasing carotenoid and chlorophyll pigment contents [26]. In *Medicago truncatula*, an R2R3-MYB TF, *MtWP1* was identified in an alfalfa flower color-isolation population, and *M**tWP1* regulates carotenoid accumulation by combining MtTT8 and MtWD40-1 [27]. These studies revealed that R2R3-MYB TFs regulate carotenoid biosynthesis. However, very little is known about the R2R3-MYB TF regulation of carotenoid metabolism in wolfberry [5].

An increasing number of *R2R3-MYB* genes were recently identified in various plant species, including Arabidopsis (*Arabidopsis thaliana*), tomato (*Solanum lycopersicum*), pepper (*Capsicum annuum*), potato (*Solanum tuberosum*), plum (*Prunus salicina*), Chinese Bayberry *(Morella rubra*), tea (*Camellia sinensis*), *Liriodendron*, pineapple (*Ananas comosus*), Chinese pistache (*Pistacia chinensis)*, and ginkgo (*Ginkgo biloba*) [28,29,30,31,32,33,34,35,36,37,38]. However, there are no reports about the *R2R3-MYB* gene family in wolfberry. Moreover, there is less information about *R2R3-MYB* than other Solanaceae genes where the *R2R3-MYB* gene family is identified, except wolfberry. This study first identified *R2R3-MYB* genes in wolfberry and performed a comprehensive analysis of the *R2R3-MYB* gene family in five Solanaceae species, including wolfberry, tomato, pepper, potato, and eggplant, to provide insights into the functional divergence among different species.

To date, genome sequences of five Solanaceae species have been sequenced and released, including wolfberry (*L. barbarum*), tomato (*S. lycopersicum*), pepper (*C. annuum*), potato (*S. tuberosum*), and eggplant (*S. melongena*) [39,40,41,42,43]. These genomic resources are informative for comparative analyses of the *R2R3-MYB* gene family among the Solanaceae species. Accordingly, this study involved genome-wide identification of *R2R3-MYB* genes in the five sequenced Solanaceae species. The evolutionary history of *R2R3-MYB* genes involved the comprehensive analysis of phylogeny, gene structure, conserved domains, and gene duplication events. Moreover, the study investigated the expression patterns of *LbaR2R3-MYB* genes by analyzing transcriptome data and quantitative-real time (qRT)-PCR analysis from fruit development stages. Furthermore, co-expression networks were constructed. The results indicate that two *LbaR2R3-MYB* genes probably regulate carotenoid metabolism. This study elucidates the evolutionary and functional roles of the *R2R3-MYB* family genes in wolfberry and other Solanaceae species.

## 2. Results

### 2.1. Identification and Sequence Analysis of R2R3-MYB Genes in Five Solanaceae Species

Members of the *R2R3-MYB* gene family were searched using two strategies: a BLASTP search using 124 AtR2R3-MYB sequences as queries and an Hidden Markov Model (HMM) search using the MYB domain file (PF00249). A total of 1326 *MYB* candidate genes were retrieved from the five Solanaceae species. The retrieved sequences were aligned to the SMART, Pfam, and CDD databases to identify R2 and R3 domains. The 716 sequences that lacked both R2 and R3 domains were removed. Thus, 610 *R2R3-MYB* genes were identified (Table 1) in wolfberry (137), tomato (133), pepper (108), potato (109), and eggplant (123) (Appendix A). 

The R2 and R3 amino acid sequences from the five Solanaceae species were used for multiple sequence alignments. In Appendix A, we observed different amino acid frequencies for each position of the R2 and R3 domains, confirming the conserved nature of these domains. All *R2R3-MYB* genes had three conserved tryptophans in the R2 domain and two in the R3 domain, where a hydrophobic amino acid replaced the first tryptophan. This observation is consistent with other studies of the *R2R3-MYB* gene family in potato, Japanese plum, and watermelon [31,32,44].

### 2.2. The Classification, Gene Structure, and Motif Composition of LbaR2R3-MYB Genes

A maximum-likelihood (ML) phylogenetic tree was constructed using full-length R2R3-MYB protein sequences from wolfberry and Arabidopsis to classify the wolfberry *R2R3-MYB* genes. This study employed Arabidopsis R2R3-MYB proteins as a reference to classify and categorize wolfberry R2R3-MYB members into 31 subgroups (designated C1 to C31) using sequence similarity (Figure 1a and Appendix A). The defined clades in Arabidopsis were labeled in the evolutionary tree. Previous studies with high bootstraps supported the largest subgroups (A1 and A3) in this study. Nevertheless, some subgroups (A7 and A31) were not retrieved from the phylogenetic tree of AtR2R3-MYB proteins. Twenty-five LbaR3-MYB proteins did not fit into any subgroup.

The exon–intron structure analysis of the 137 *LbaR2R3-MYB* genes indicated that introns disrupted most of their coding sequences, except for one gene from subgroup A25 and three from subgroup A26 (Figure 1b). The number of exons in *LbaR2R3-MYB* genes ranged from one to twelve, with an average of 3.0. Among these, 85 *LbaR2R3-MYB* genes had three exons, accounting for approximately 62% of the *LbaR2R3-MYB* gene family, whereas 14% of the *LbaR2R3-MYB* genes had more than three exons. Most *LbaR2R3-MYB* genes clustered in related groups with similar exon-intron structures, such as A1, A6, A9, A10, A21, and A22 (Figure 1b). 

Subsequently, the MEME program identified 20 conserved motifs among wolfberry R2R3-MYB proteins (Appendix A). Most of the *LbaR2R3*-*MYB* DNA-binding domains contained motifs 1, 2, 3, 4, 5, and 8 (Figure 1c). The R2R3-MYB domain is highly conserved; thus, R2R3-MYB members within the same subgroup usually have similar motif composition, but different subgroups vary greatly. Moreover, some subgroup-specific motifs were detected, probably required for subgroup-specific functions. For instance, motifs 17 and 18 were only found among subgroup A13, whereas motif 10 was unique to subgroup A31. These results indicate the divergence of *LbaR2R3*-*MYB* TFs.

### 2.3. Comparative Phylogenetic Analysis of the R2R3-MYB Family in Five Solanaceae Species

A maximum-likelihood phylogenetic tree was constructed using all R2R3-MYB protein sequences from Arabidopsis and the five Solanaceae species. Figure 2 and Appendix A show the condensed and complete phylogenetic trees. The comparative phylogenetic analysis divided R2R3-MYB proteins from the six species into 36 subgroups (designed as C1 to C36). The tree topology showed that only subgroup C3 included R2R3-MYB proteins of all five Solanaceae species. Meanwhile, subgroup C35 included members from all Solanaceae species except wolfberry. There were no wolfberry-specific subgroups. Particularly, subgroup C18 was only present in Arabidopsis and not in Solanaceae species.

Most clades included members from all six species, but the R2R3-MYB proteins were not equally represented in the six species within any given clade. Some subgroups (C4, C5, C6, C13, and C19) contained more abundant R2R3-MYB from Solanaceae species than from Arabidopsis, while several subgroups (C32 and C33) contained fewer Solanaceae R2R3-MYB proteins. Moreover, many clades from wolfberry had more R2R-MYB proteins than the other four Solanaceae species.

### 2.4. Analysis of Gene Duplication Events and Chromosomal Distributions of the R2R3-MYB Gene Family

Multigene families originate from gene duplication and are the proven prominent feature of plant genome evolution [45]. Five modes of gene duplication, including whole-genome duplication (WGD), tandem duplication (TD), proximal duplication (PD), transposed duplication (TRD), and dispersed duplication (DSD), were analyzed in the five Solanaceae species to investigate the origin of the *R2R3-MYB* family genes.

The study investigated different gene duplication events and identified their contributions to expanding the *R2R3-MYB* gene family. There were 842 duplicated gene pairs in the five Solanaceae species. DSDs (358 gene pairs), WGDs (213 gene pairs), and TRDs (159 gene pairs) were the maximum number of gene pairs, suggesting that the expansion of the R2R3-MYB gene family was mainly associated with DSD, WGD, and TRD events. In contrast, only 49 and 72 PDs and TDs were identified in the R2R3-MYB gene family. The number of WGD-pairs in wolfberry (63), tomato (54), and eggplant (51), which shared a recent lineage-specific WGD event, are greater than potato (21) and pepper (24). The disparity indicates the importance of WGD events in the R2R3-MYB family expansion in wolfberry, tomato, and eggplant. DSD and TRD events occurred more frequently in pepper, which had not experienced recent WGD events, suggesting the importance of single-gene duplications in expanding the R2R3-MYB family during the long-term evolution of these genomes (Figure 3 and Appendix A).

The study also analyzed the distribution of *R2R3-MYB* genes on the chromosomes of five Solanaceae species. For wolfberry, 137 *R2R3-MYB* genes were randomly distributed on 12 chromosomes. The wolfberry chromosome 1 had the highest number of genes (21) compared with the other chromosomes. However, chromosome 10 had only four *R2R3-MYB* genes. There was no significant correlation between the chromosome length and the number of *LbaR2R3-MYB* genes (Appendix A). Similarly, the *R2R3-MYB* genes were randomly distributed in the other four Solanaceae species (Figure 4). The study further identified intra-genomic synteny blocks for each species. There were 153 syntenic gene pairs among the five Solanaceae species (Figure 4). Of these, wolfberry (Figure 4a), tomato (Figure 4b), pepper (Figure 4c), potato (Figure 4d), and eggplant had 48, 38, 16, 16, and 35 syntenic pairs, respectively (Figure 4e and Appendix A).

The comparative syntenic map of wolfberry associated with Arabidopsis and four other Solanaceae species was constructed. The study also identified the orthologous *R2R3-MYB* genes to infer the evolutionary mechanisms of *LbaR2R3-MYB* genes (Figure 5). A total of 80, 26, 77, 87, and 17 orthologous gene pairs were identified between wolfberry and tomato, pepper, potato, eggplant, and Arabidopsis, respectively. Interestingly, some collinear gene pairs were only found between wolfberry and specific species. For example, the collinear gene pairs *Lba08g01691-Solyc10g083900.2.1* and *Lba11g00940-Capana07g001606* were only available between wolfberry and tomato and between wolfberry and pepper, respectively. Forty-one *LbaR2R3-MYB* genes had collinear relationships with other selected four Solanaceae species. However, 79 *LbaR2R3-MYB* genes were associated with Solanaceae-specific collinear gene pairs but absent between wolfberry and Arabidopsis (Appendix A). The formation of these species-specific collinear gene pairs might be related to the evolutionary mechanism in Solanaceae species. Additionally, some *LbaR2R3-MYB* genes were associated with two or more orthologous gene pairs. For example, *Lba02g02412* is orthologous to *Solyc04g078420.1.1*, *PGSC0003DMT400008569*, and *Smechr0202641.1.*

### 2.5. Nonsynonymous (Ka) and Synonymous (Ks) Substitutions per Site and Ka/Ks Analysis of the R2R3-MYB Family Genes

The Ks value estimates the evolutionary history of WGD events. The mean Ks values of WGD-derived gene pairs in wolfberry, tomato, pepper, potato, and eggplant were 1.14, 1.11, 1.36, 1.35, and 1.35, respectively. Lower Ks values of WGD-derived gene pairs in the five Solanaceae species suggested that the genes were duplicated and retained from recent WGD events (Appendix A). The Ka/Ks ratios of the duplicated gene pairs in wolfberry, tomato, pepper, potato, and eggplant were <1, indicating that *R2R-MYB* genes evolved under strong purifying selection. However, five gene pairs (*Lba09g01300* and *Lba09g01301* (Ka/Ks ~2.55), *Lba06g00593* and *Lba03g02818* (Ka/Ks ~1.31), *PGSC0003DMT400015155* and *PGSC0003DMT400015156* (Ka/Ks ~1.59), *SmeSca00628.1* and *SmeSca00696.1* (Ka/Ks ~1.05), and *SmeSca00639.1* and *SmeSca00696.1* (Ka/Ks ~2.11) in wolfberry, potato, and eggplant) had higher Ka/Ks ratios, suggesting a complicated evolutionary history. For wolfberry, the mean Ka/Ks values for WGD, TD, PD, TRD, and DSD gene pairs were 0.24, 0.54, 0.61, 0.31, and 0.30, respectively (Figure 6 and Appendix A). The PD gene pairs had the highest Ka/Ks ratio compared with other types of duplicated gene pairs, indicating that PD evolved at a higher rate than the other gene pairs (Figure 6).

### 2.6. Expression of Carotenoid-Biosynthetic Genes and R2R3-MYB DEGs in Wolfberry

RNA-seq data determined the expression profiles of *LbaR2R3-MYB* differentially expressed genes (DEGs) and carotenoid biosynthesis genes (CBGs) in RF fruits at five development stages: 12 (S1), 19 (S2), 25 (S3), 30 (S4), and 37 (S5) days after full bloom (DAF). A total of 45 (32.8%) DEGs of the *LbaR2R3-MYB* gene family (adjust *p*-value < 0.01, |log2-fold change| > 1) were identified and expressed in five developmental stages (Appendix A). The expression values clustered the *LbaR2R3-MYB* DEGs into four main groups, I to IV (Figure 7a). Most genes in group I were highly expressed in all the fruit development stages. Except for two genes (*Lba05g00371* and *Lba05g00389*): group II had a lower expression, while group III contained five *LbaR2R3-MYB* genes, which were highly expressed at 12 DAF and 19 DAF. The expression levels of ten *LbaR2R3-MYB* genes increased gradually with fruit development. These results suggest that *LbaR2R3-MYB* genes regulate wolfberry fruit development. 

The study also analyzed the expression levels of CBGs in the carotenoid biosynthesis pathway (Figure 7b). Genes *BCH1*, *PSY1*, *PDS*, *ZDS*, *ZISO*, and *CYP97A* were highly expressed in the late stages (25 DAF to 37 DAF) of fruit development (Appendix A).

### 2.7. Co-Expression Analysis of Carotenoid-Biosynthetic Gene and LbaR2R3-MYBs

A co-expression network of carotenoid biosynthetic genes was constructed to investigate the potential of *LbaR2R3-MYB* TFs for regulating carotenoid biosynthesis in wolfberry (Figure 8). First, the expression levels of 45 *LbaR2R3-MYB* DEGs and 15 CBGs were used to calculate Pearson’s correlation coefficients (PCCs). Gene pairs with |PCCs| < 0.80 and *p* > 0.05 were removed, and the remaining gene pairs were used to construct the co-expression network. From the network, 15 CBGs, 32 *LbaR2R3-MYB* TFs, and 230 pairs were constructed a co-expression relationship (Appendix A). Several LbaR2R3-MYB TFs (8, 9, 8, 9, and 7) had a highly positive correlation with *PSY1*, *BCH1*, *ZDS*, *PDS*, and *ZISO* genes, respectively, whereas some of LbaR2R3-MYB TFs (11, 8, 12, 11, 13) had a highly negative with *PSY1*, *BCH1*, *ZDS*, *PDS*, and *ZISO* genes, respectively. These TFs might regulate the expression of five important genes for accumulating carotenoids. For example, gene *BCH1* positively correlated with nine TFs and negatively correlated with eight TFs. The coefficients of eight positively correlated TFs were >0.900, while eight negatively correlated TFs were <−0.900 (Appendix A). 

### 2.8. Gene Expression Analyses with qRT-PCR

Seven candidate-*L**baR2R3-MYB* genes were selected for qRT-PCR validation. The results indicate that the expression of three *LbaR2R3-MYB* genes (*Lba11g01830*, *Lba05g01910*, and *Lba02g01219*) highly correlated with the content of total carotenoids during RF fruit development (Figure 9a). The expression of *Lba11g01830* decreased from S1 (12 DAF) to S2 (19 DAF) and increased sharply from S3 (25 DAF) to S5 (37 DAF) (Figure 9b). Surprisingly, the relative expression, RNA-seq data, and the changes in total carotenoid contents correlated with *Lba05g01910* and *Lba02g01219* from S3 (25 DAF) to S5 (37 DAF) (Figure 9c,d).

We constructed a maximum-likelihood tree from three wolfberry *R2R3-MYB* genes and seven characterized R2R3-MYB genes from other species. Two *LbaR2R3-MYB* genes (*Lba11g01830* and *Lba02g01219*) shared high sequence identity with the reported function-known *R2R3-MYB* genes (Figure 10). Altogether, we speculate that these two *LbaR2R3-MYB* genes are important in carotenoid biosynthesis.

## 3. Discussion

The *R2R3-MYB* gene family is among the largest families in plants. To date, members of the *R2R3-MYB* gene family have been identified and analyzed in different land plant species, including four Solanaceae species: tomato, potato, pepper, and eggplant. The number and composition of the *R2R3-MYB* gene family differ in different plants [29,34,35]. Ancient polyploidy events (also known as WGDs) and additional recent lineage-specific WGDs have presumably caused varying numbers of *R2R3-MYB* genes within land plants [46]. A recent study sequenced and released the genome of the wolfberry (*Lycium bararum* L.), an economically important genus of the Solanaceae family [39]. However, this study identifies the first *R2R3-MYB* gene family from the wolfberry genome and reports a comparative analysis of the *R2R3-MYB* gene family in five Solanaceae species. The size of the *R2R3-MYB* gene family is diverse in the five Solanaceae genomes. Surprisingly, the number of *R2R3-MYB* genes in wolfberry (137) and tomato (133) is greater than pepper (108) and potato (109) (Table 1), suggesting that pepper, potato, and eggplant experienced more frequent gene losses. Wolfberry and tomato probably experienced lineage-specific WGD, while pepper, potato, and eggplant did not. Therefore, this recent WGD event likely generated different *R2R3-MYB* gene numbers in the investigated Solanaceae species.

Phylogenetic tree analysis displayed 610 R2R3-MYB proteins from the six analyzed species categorized into 36 subgroups (C1–C36). The 610 R2R3-MYB proteins included 137, 133, 108, 109, 123, and 124 proteins from wolfberry, tomato, pepper, eggplant, and Arabidopsis. Only Solanaceae contained the five species-specific subgroups (C1, C3, C34, C35, and C36), suggesting their species-specific role in Solanaceae. Altogether, these results indicate that wolfberry, tomato, pepper, potato, and eggplant are closely related to Arabidopsis. Subgroup C18 only contained Arabidopsis R2R3-MYB proteins, further establishing that the corresponding S12 subgroup of Arabidopsis was the Arabidopsis-specific subfamily regulating glucosinolates biosynthesis [28,47]. This subgroup C18 of the phylogenetic tree was similar to other species [48]. The unequal representation of R2R3-MYB proteins within the divided subgroups suggested that *R2R3-MYB* gene expansion events may be more active in certain plant species.

Gene duplication is a major source of new genes in evolution that involves whole genome/segmental duplication (WGD/SD), TD, PD, TRD, and DSD. Gene duplication is crucial for gene family expansion and evolution. For example, DSD and WGD events expanded the *ADH*, *COMT*, and *SWEEET* gene families [49,50,51], whereas TD events expanded the *HSP* gene family [52]. The present study showed that DSD, WGD, and TRD significantly expanded the *R2R3-MYB* gene family in the five Solanaceae species. Moreover, the Ka, Ks, and Ka/Ks analyses showed that the mean Ks values of WDG-derived gene pairs were much lower in wolfberry and tomato than pepper, potato, and eggplant. This observation supports a lineage-specific WGD event (~37 MYA) shared by wolfberry and tomato. Additionally, wolfberry TRD-pairs had a higher Ka/Ks ratio, indicating that TRD-derived *R2R3-MYB* genes experienced a rapid functional divergence.

Plant R2R3-MYB control diverse pathways, such as secondary metabolism (including the carotenoid biosynthesis pathway), plant growth and development, biotic and abiotic stresses [53]. For instance, tomato stamens and pistils predominately express the tomato *SlMYB33* gene, which regulates tomato flowering and pollen maturity. *SlMYB75* positively regulates the accumulation of anthocyanins by transcriptionally regulating downstream genes [54]. Meanwhile, *SlMYB102* participates in stress tolerance by regulating several molecular and physiological processes [55]. Despite the diverse functions of the *R2R3-MYB* gene family, this study focused on the roles of *R2R3-MYB* in regulating carotenoid biosynthesis. The RNA-seq analysis identified *45 LbaR2R3-MYB* genes and 15 carotenoid biosynthetic genes at various expression levels in the five stages of wolfberry fruit development. Subsequently, we constructed a co-expression network of carotenoid regulation. A comprehensive analysis of transcriptome and co-expression identified seven *LbaR2R3-MYB* genes that were validated by qRT-PCR. Previous studies demonstrated that the seven R2R3-MYB TFs regulate carotenoid biosynthesis [23,24,25,26,27]. Indeed, the qRT-PCR expression patterns of two *LbaR2R3-MYB* genes (*Lba11g01830* and *Lba02g01219*) were consistent with the carotenoid accumulation trend in fruit development, indicating that these two genes may regulate carotenoid synthesis. 

Therefore, a phylogenetic tree was constructed using full-length amino acids from the seven R2R3-MYB TFs and the three candidate-LbaR2R3-MYB TFs (Lba11g01830, Lba05g01910, and Lba02g01219). Lba11g01830 and Lba02g01219 were highly sequence identity with CrMYB68 and AdMYB7 proteins, indicating that these two *LbaR2R3-MYB* genes regulate carotenoid biosynthesis in wolfberry. Whether these R2R3-MYB TFs bind to promoter sequences of carotenoid biosynthetic genes (*PSY*, *PDS*, and *ZDS*) needs further analysis.

## 4. Materials and Methods

### 4.1. Plant Materials

All experimental materials were collected from the wolfberry germplasm of the Center of Wolfberry Engineering Technology Research, Yinchuan, Ningxia (38°38′49″ N, 106°9′10″ E), China. The fruits of *L. chinense* var. potaninii (RF) were collected at five different developmental stages, 12 (S1), 19 (S2), 25 (S3), 30 (S4), and 37 (S5) days after full bloom (DAF) (Appendix A). All samples were ground in liquid nitrogen and stored at −80 ℃ for further study.

### 4.2. Identification and Sequencing of R2R3-MYB Genes in Five Solanaceae Species

The sequence of 124 AtR2R3-MYB proteins was retrieved from the Arabidopsis Information Resource (https://www.arabidopsis.org/, accessed on 2 March 2021) to identify *R2R3-MYB* TF family genes. The genome sequence of wolfberry (*Lycium barbarum*) and genome annotation files were downloaded from the NCBI database (https://www.ncbi.nlm.nih.gov/, accessed on 3 July 2021) with accession number PRJNA640228 [32]. However, genome sequences of tomato (*Solanum lycopersicum*) and potato (*Solanum tuberosum*) were downloaded from the Solanaceae Genomics Network (https://solgenomics.net/, accessed on 2 July 2021). The genome sequence of *Capsicum annuum*-Zunla-1 was downloaded from the China National Genebank (https://db.cngb.org/cnsa/, accessed on 2 July 2021). Additionally, the genome sequence of eggplant ‘HQ-1315’ (*Solanum melongena*) was downloaded from the Eggplant Genome Database (http://eggplant-hq.cn/Eggplant/home/index, accessed on 2 July 2021).

The study used two strategies to search for candidate *R2R3-MYB* genes. First, sequences of 124 AtR2R3-MYB proteins were used as queries for BLASTP searches against local protein databases of the Solanaceae species with E-values < 1 × 10^−10^. Subsequently, the MYB domain (PF00249) obtained from the Pfam database (http://pfam.xfam.org/, accessed on 5 July 2021) was used to construct a hidden Markov model (HMM) for searching against protein databases with E-values < 1 × 10^−10^ using the HMMER v3.3.2 software [56]. Finally, three databases, including SMART (http://smart.embl-heidelberg.de/, accessed on 5 July 2021), Pfam (http://pfam.xfam.org/, accessed on 5 July 2021), and CD search (https://www.ncbi.nlm.nih.gov/Structure/cdd/wrpsb.cgi, accessed on 5 July 2021) confirmed the presence of the R2R3 domain [57,58,59]. The protein sequences without the R2R3 domain and redundant sequences were manually removed. The ProtParam tool (https://web.expasy.org/protparam/, accessed on 10 July 2021) predicted the isoelectric point (pI) and molecular weight (*M*_W_) of all R2R3-MYB proteins based on their deduced amino acid sequences. 

### 4.3. Conserved Motif Analysis of R2R3-MYB Genes in Wolfberry

The Gene Structure Display Server (http://gsds.gao-lab.org/index.php, accessed on 10 August 2021) [41] graphically displayed the exon–intron organizations of the *LbaR2R3-MYB* genes using Generic Feature Format Version3 (GFF3) annotation files of *LbaR2R3-MYB* genes. The MEME suite (https://meme-suite.org/meme/, accessed on 10 August 2021) [60] predicted the conserved motifs of *LbaR2R3-MYB* genes using the following parameters: maximum numbers of different motifs, 20; minimum motif width, 6; and maximum motif width, 50. The results were visualized using iTOL (https://itol.embl.de/, accessed on 10 August 2021) [61].

### 4.4. Conserved Motif Analysis of R2R3-MYB Genes in Wolfberry

Complete amino acid sequences of wolfberry, *Arabidopsis thaliana*, and the other four Solanaceae species were aligned using the Muscle program with default parameters [62]. Then, a maximum-likelihood (ML) phylogenetic tree was constructed using IQ-TREE [63]. The best-fit substitution model, JTT+G, was determined by MEGA 6.06 [64] and incorporated in the IQ-TREE with 1000 bootstraps. OrthoFinder [65] constructed the taxonomy tree of the five Solanaceae species, which was visualized using the iTOL [61] online tool and Figtree v1.4.4 (https://tree.bio.ed.ac.uk/software/figtree/, accessed on 10 August 2021).

### 4.5. Identification of Gene Duplications, Chromosomal Location, and Collinearity Analysis

The DupGen_finder pipeline [66] further identified paralogous *R2R3-MYB* gene pairs derived from WGD, TD, PD, TRD, and DSD. Briefly, *Arabidopsis thaliana* was selected as the outgroup to identify duplicated gene pairs. Then, all of the species were subjected to a BLASTP search against *Arabidopsis thaliana*. The simplified Gff3 files were generated using Tbtools [67].

The genome annotation files provided the information on the chromosomal locations of the *R2R3-MYB* genes in *Lycium barbarum*, *Solanum lycopersicum*, *Capsicum annuum*, *Solanum tuberosum*, and *Solanum melongena*. Tbtools [67] analyzed collinear relationships between wolfberry and the other five plant species. First, the BLASTP algorithm detected potential homologous gene pairs between wolfberry and the other five plant species. Second, the BLASTP results and the gene location information were uploaded to Tbtools to identify and visualize syntenic chains. The syntenic gene pairs within each species were also identified following the above steps. Genes located on unanchored scaffolds were not included.

### 4.6. Ka and Ks Calculation

The nonsynonymous (Ka) and synonymous substitution rates (Ks) of syntenic gene pairs were calculated using Tbtools with the Nei–Gojobori (NG) method [67]. Briefly, the coding sequences and duplicated gene pairs were prepared first. The two files were deposited into Tbtools to acquire readable results, including Ka, Ks, Ka/Ks, and *p*-value.

### 4.7. Expression Profiling of LbaR2R3-MYB Genes with RNA-seq

The raw RNA-seq reads were deposited in the NCBI database. The adapter sequences, low-quality reads (quality score < 15), and poly (A/T) tails were removed from raw reads using fastp [68]. The Hisat2 [69] software aligned clean reads to the reference genome and feature counts estimated transcript abundances. The fragments per kilobase million (FPKM) measured the expression levels of the *R2R3-MYB* genes. The expression level of each *R2R3-MYB* gene was displayed in a heatmap using the R software (https://www.r-project.org/, accessed on 3 September 2021).

### 4.8. Quantitative Real-Time PCR Analysis

Total RNA was extracted from the fruits of two wolfberry cultivars using the TRNzol Universal Reagent (TIANGEN, Beijing, China) following the manufacturer’s instructions. The RNA was resolved on 1% agarose gel for quality assessment and quantified using Nanodrop one (Nanodrop Technologies, Wilmington, DE, USA). Genomic DNA was eliminated, and first-strand cDNA was synthesized using the Easycript one-step RT-PCR supermix (Transgen Bio, Inc., Beijing, China). The qRT-PCR was conducted on a CFX96 TouchTM Real-Time PCR Detection System (Bio-Rad, Hercules, CA, USA) using the perfectStartTM Green qPCR Supermix (Transgen Bio, Inc., Beijing, China). The qPCR conditions were as follows: 30 s at 95 °C, followed by 40 cycles of 5 s at 95 °C, 30 s at 60 °C, and 65–95 °C melting curve detection. Standard curve analysis of serially diluted cDNA estimated the qPCR efficiency, and *LbACTIN* was used for template normalization [5]. The relative abundance was calculated using the comparative Ct (2^−ΔΔCt^) method. All qRT-PCR primers were designed in Primer5.0 and listed in Appendix A.

### 4.9. Total Carotenoid Extraction and Measurement

The extraction and determination of total carotenoids were performed as previously described [70], with some modifications. Briefly, ~5 mg of fresh fruits were ground into fine powder in liquid nitrogen and extracted three times using 150 mL tetrahydrofuran with 0.1% butylated hydroxytoluene (BHT) via ultrasonic treatment for 45 min at 7 W. After centrifugation (4000× *g* for 10 min at 4 °C), the extracts were combined into a 50 mL tube and mixed by shaking with 5 mL NaCl-saturated solution for 1 min, and the supernatant was collected. The petroleum ether extraction solution was merged and condensed by vacuum rotary evaporation at 35 °C. The concentrated carotenoids residue was dissolved using methylene chloride. The absorbance of the solution against the black at 450 nm was determined using UV-vis spectrophotometry (UV-1800, Shimadzu Co., Ltd., Kyoto, Japan).

### 4.10. Construction of Co-Expression Network

The co-expression network was constructed based on RNA-seq data and carotenoid contents to investigate the regulatory network between structural genes of carotenoid biosynthesis and R2R3-MYB TFs. First, Pearson’s correlation coefficients (PCCs) were calculated to select the positive and negative correlations between the structural genes and R2R3-MYB TFs. PCC values < 0.8 were removed, and the networks were visualized in Cytoscape v3.8.2 [71]. 

### 4.11. Statistical Analysis

The experiments involved three biological replicates. Statistical significance (Student’s *t*-tests) was analyzed using the R software, and a *p <* 0.05 was considered statistically significant.

## 5. Conclusions

In the present study, genome-wide identification and bioinformatic analyses of *R2R3-MYB* genes in wolfberry (*Lycium barbarum* L.) and four other Solanaceae Species were performed. A total of 610 homologous *R2R3-MYB* genes were identified. Among them, 137 belonged to wolfberry. The *R2R3-MYB* genes are divided into 36 large clades following the classification results from model plants. DSD, WGD, and TRD were the primary forces driving the *R2R3-MYB* gene family expansion. Purifying selection was the main evolutionary force on *R2R3-MYB* genes except for a gene pair with Ka/Ks values >1. In addition, integrated bioinformatics analysis and experimental verification identified two candidate-*LbaR2R3-MYB* genes (*Lba11g01830* and *Lba02g01219*) related to carotenoids accumulation. These results provide insights into the evolutionary history and a foundation for understanding the molecular mechanisms underlying carotenoid biosynthesis.

## Figures and Tables

**Figure 1 ijms-23-02259-f001:**
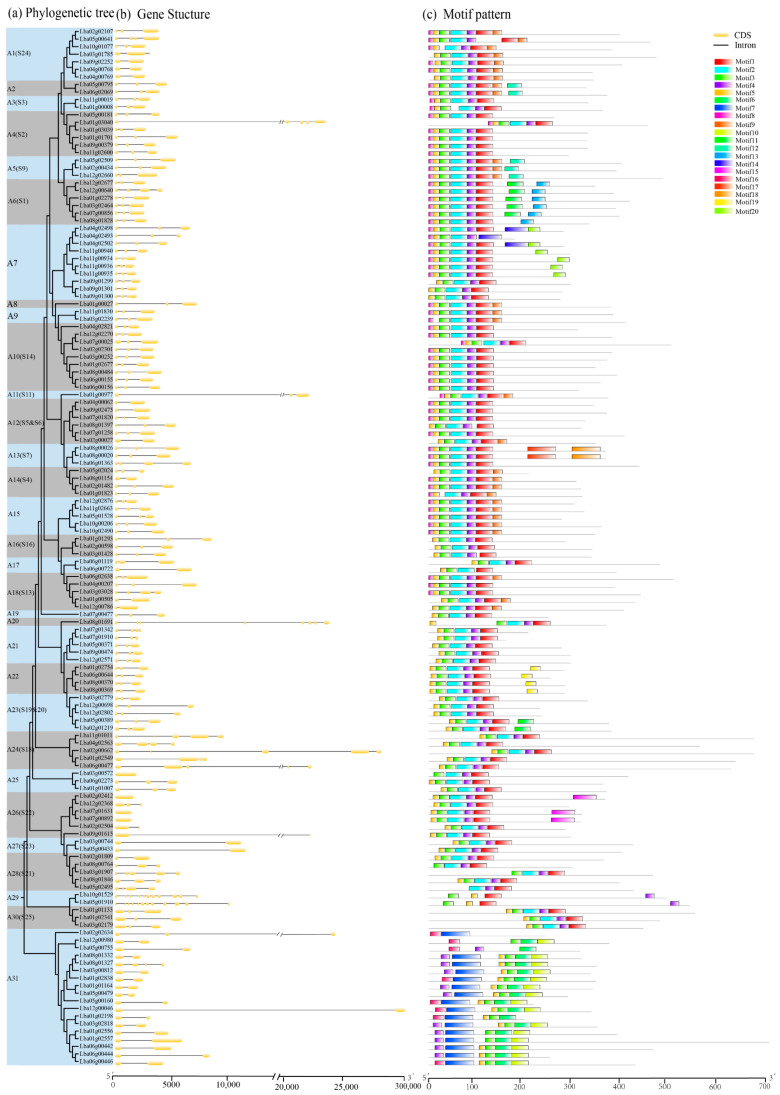
Phylogenetic relationship, conserved protein motifs, and gene structure in *LbaR2R3-MYB* genes. (**a**) The maximum-likelihood (ML) phylogeny includes 137 R2R3-MYB proteins from wolfberry, grouped into 31 subgroups, sequentially designated as C1 to C31. The corresponding MYB subgroup names in Arabidopsis are also marked. (**b**) Gene structure of wolfberry *R2R3-MYB* genes. Yellow boxes indicate exons; black lines indicate introns. (**c**) The motif composition of wolfberry R2R3-MYB proteins. Twenty different motifs are displayed in different colored boxes. The length of proteins can be estimated using the scale at the bottom.

**Figure 2 ijms-23-02259-f002:**
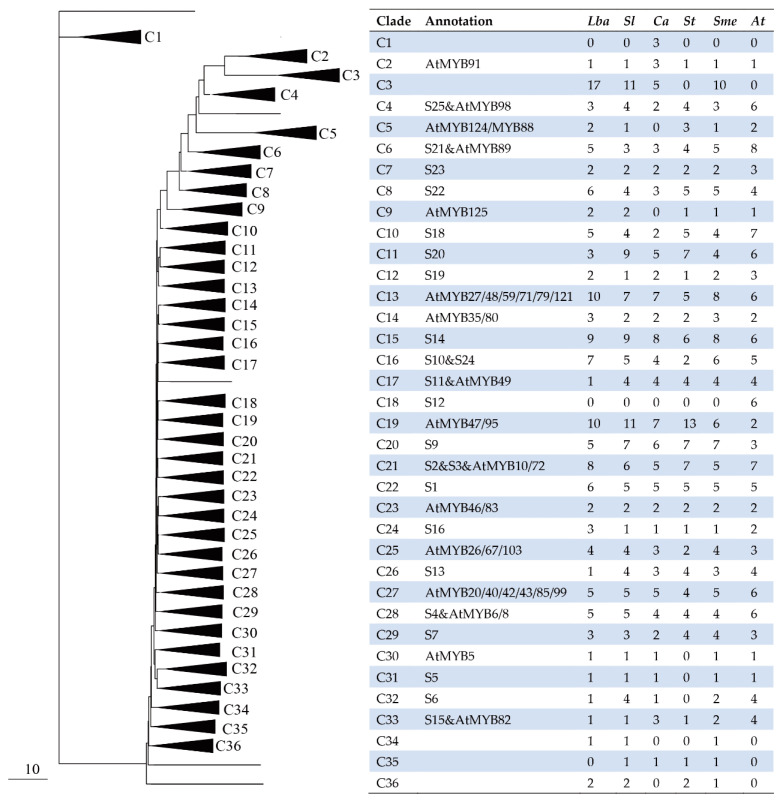
A phylogenetic tree of R2R3-MYB proteins. A total of 137 proteins from wolfberry (*Lba*), 133 from tomato (*Sl*), 108 from pepper (*Ca*), 109 from potato (*St*), 123 from eggplant (*Sme*), and 124 from Arabidopsis (At) were used. The full-length amino acid sequences of R2R3-MYB proteins were aligned using Muscle and the phylogenetic tree was constructed using the maximum-likelihood method. R2R3-MYB proteins from the six species clustered into 36 subgroups (triangles) designated as C1 to C36. Four proteins did not fit well into subgroups (lines). The tables on the right indicate the number of subgroup members in each species. The uncompressed tree is available in Appendix A.

**Figure 3 ijms-23-02259-f003:**
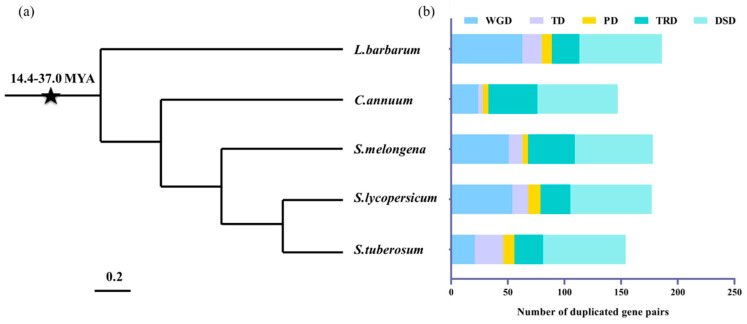
The number of R2R3-MYB gene pairs derived from different gene duplication events in the five Solanaceae species. (**a**) The phylogenetic relationship among the five Solanaceae species. (**b**) The number of different models of duplicated gene pairs in each species. The x-axis represents the number of duplicated gene pairs. The y-axis represents species. Whole-genome duplication (WGD), tandem duplication (TD), proximal duplication (PD), transposed duplication (TRD), and dispersed duplication (DSD).

**Figure 4 ijms-23-02259-f004:**
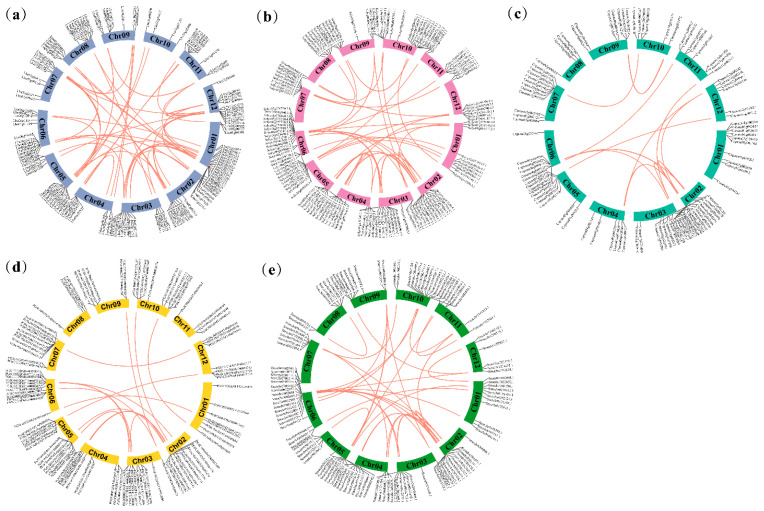
Gene location and collinearity analysis of the R2R3-MYB gene family. (**a**) Wolfberry; (**b**) tomato; (**c**) pepper; (**d**) potato; (**e**) eggplant. The *R2R3-MYB* genes in five Solanaceae species mapped on the different chromosomes. Red-colored lines joined gene pairs with a syntenic relationship.

**Figure 5 ijms-23-02259-f005:**
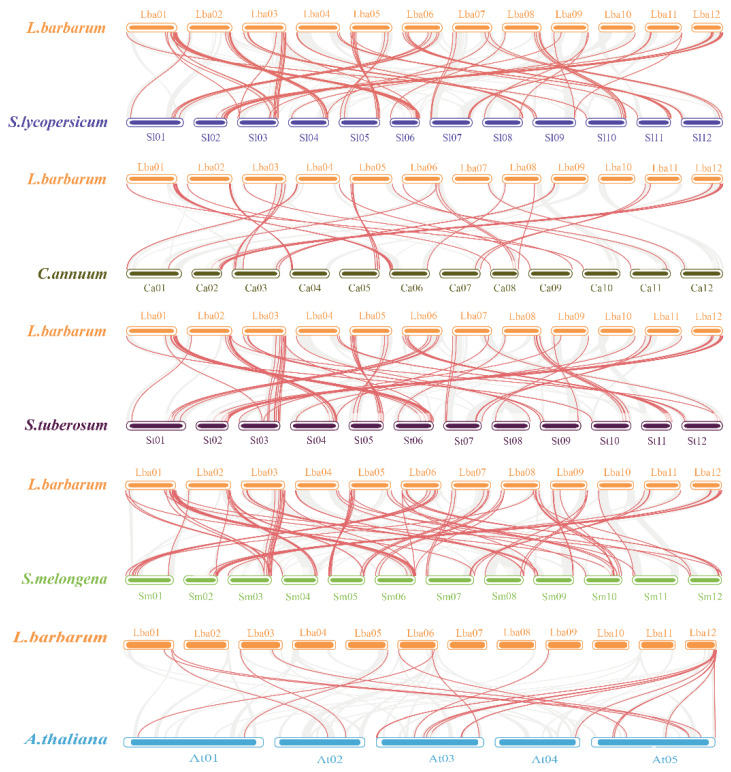
Synteny analyses of *R2R3-MYB* genes between wolfberry and the five representative species. The gray lines in the background indicate the collinear block with wolfberry and other five plant species genomes, while red lines highlight syntenic *R2R3-MYB* gene pairs, respectively.

**Figure 6 ijms-23-02259-f006:**
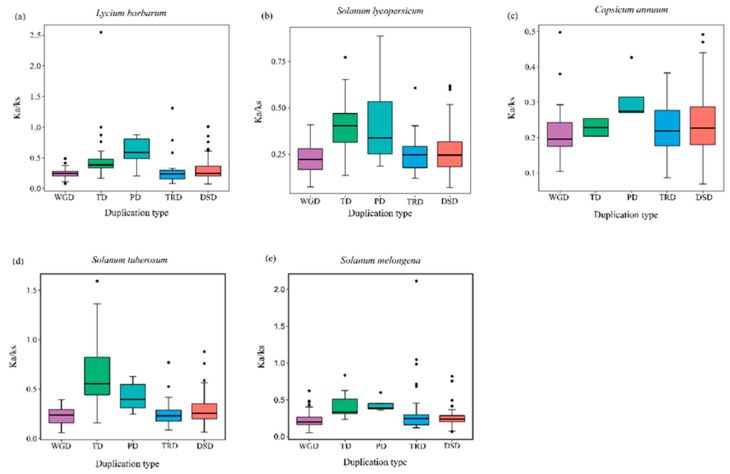
Ka/Ks ratios of five Solanaceae species. The x-axis represents five different duplication types. WGD: whole-genome duplicates; TD: tandem duplicates; PD: proximal duplicates; TRD: transposed duplicates; DSD: dispersed duplicates. The y-axis indicates the Ka/Ks ratio. (**a**) Wolfberry; (**b**) tomato; (**c**) pepper; (**d**) potato; (**e**) eggplant.

**Figure 7 ijms-23-02259-f007:**
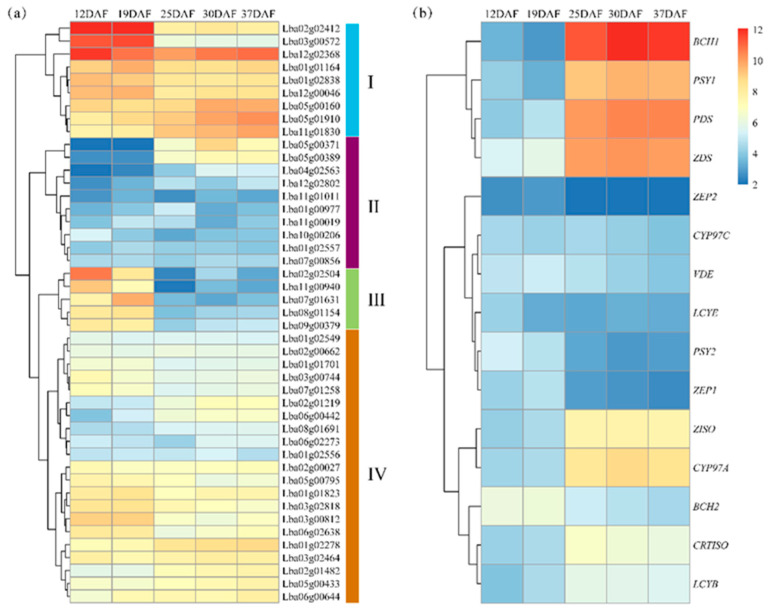
A heatmap of *LbaR2R3-MYB* DEGs and CBGs in wolfberry. Expression profiles using RNA-seq Fragments Per Kilobase Million (FPKM) data in fruit development between 12 DAP and 37 DAP. (**a**) *LbaR2R3-MYB* DEGs expression level. (**b**) CBGs, including *PSY:* phytoene synthase; *PDS*: phytoene desaturase; *ZDS*: ζ-carotene desaturase; *ZISO*: 15-cis-ζ-carotene isomerase; *CRTISO*: carotenoid isomerase; *LCYB*: lycopene β-cyclase; *LCYE*: lycopene ε-cyclase; *BCH*: β-carotene hydroxylase; *CYP97A*: cytochrome P450-type β-hydroxylase; *CYP97C*: cytochrome P450-type monooxygenase; *ZEP*: zeaxanthin epoxidase; and *VDE*: violaxanthin de-epoxidase. Log2 (FPKM +1) values were displayed according to the color code (Top right). The red and blue colors represent the highest and lowest expression levels, respectively.

**Figure 8 ijms-23-02259-f008:**
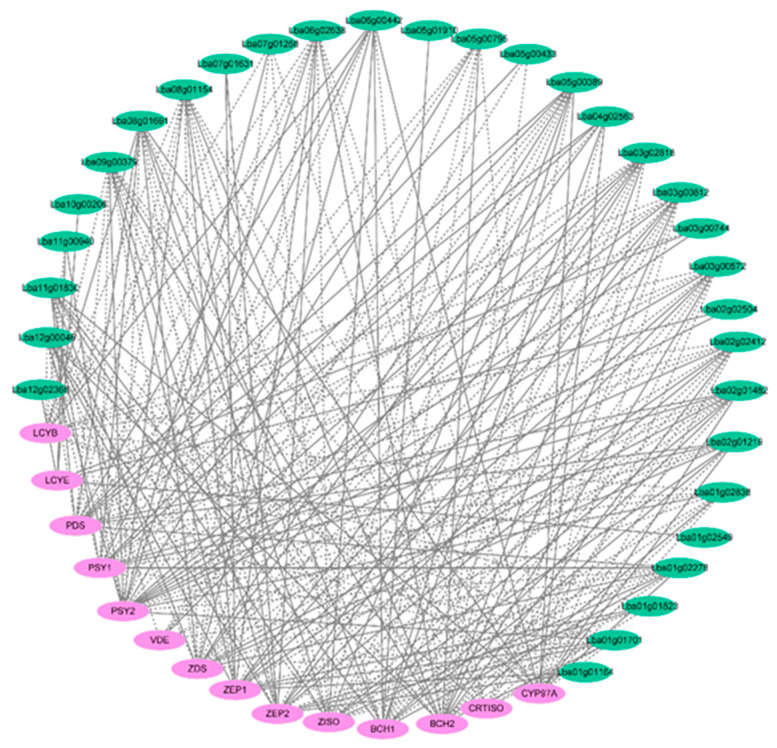
Co-expression networks of *LbaR2R3-MYB* DEGs and CBGs. Green boxes represent *LbaR2R3-MYB* DEGs, and purple boxes represent CBGs. The solid lines indicate positive correlations, and the dotted lines indicate negative correlations.

**Figure 9 ijms-23-02259-f009:**
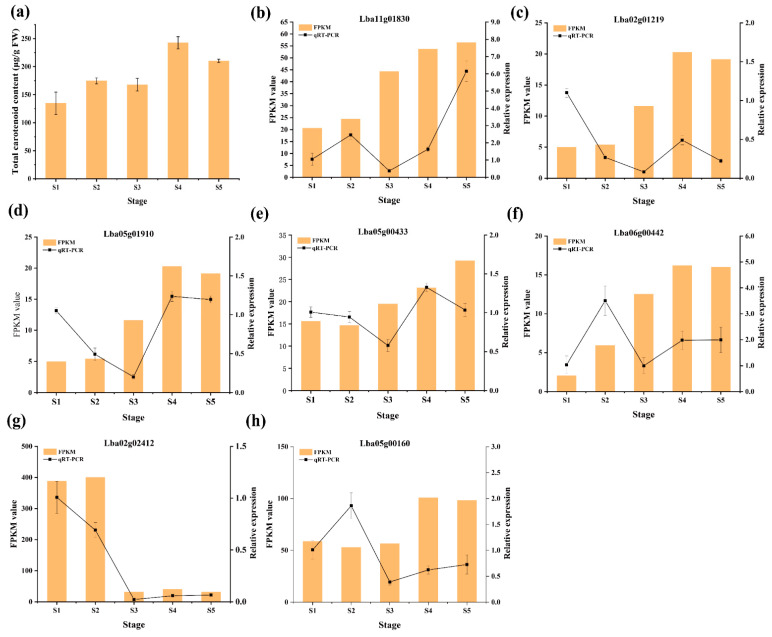
The relative expression levels of seven *LbaR2R3-MYB* genes at different stages. The *x*-axis indicates the five distinct periods. The y-axis indicates the relative expression and FPKM value. Data are presented as mean ± SDs (*n = 3*). (**a**) Carotenoid contents; (**b**) *Lba11g01830*; (**c**) *Lba02g01219*; (**d**) *Lba05g01910*; (**e**) *Lba05g00433*; (**f**) *Lba06g00442*; (**g**) *Lba02g02412*; (**h**) *Lba05g00160*.

**Figure 10 ijms-23-02259-f010:**
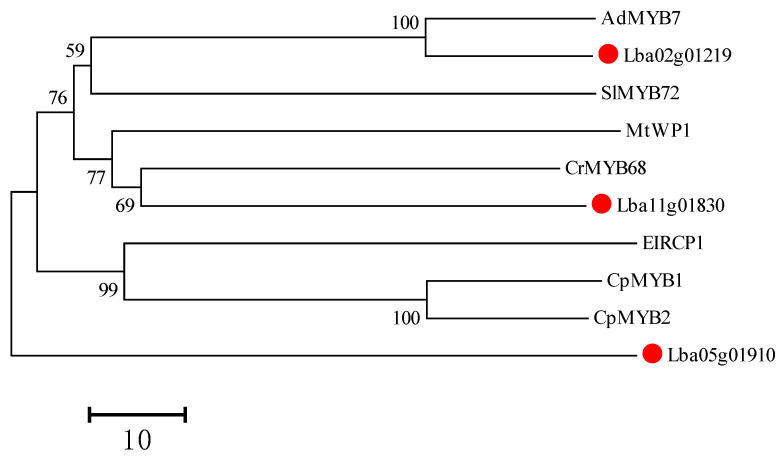
Phylogenetic analysis of candidate and seven function-known *R2R3-MYB* genes from other species. The maximum-likelihood method was used to construct the phylogenetic tree. The Genebank accession numbers are as follows: *AdMYB7* (AXP34749.1), *CrMYB68* (ASK51185.2), *SIMYB72* (Solyc07g055000), *EIRCP1* (KR053165.1), *CpMYB1*(XP_021903563.1), and *Mt**WP1*(Medtr0197s0010). Three candidate-LbaR2R3-MYB genes (*Lba02g01219*, Lba11g01830, and *Lba05g01910*) are marked with a red dot.

**Table 1 ijms-23-02259-t001:** Genomic information and identified *R2R3-MYB* gene numbers in five Solanaceae species.

Common Name	Scientific Name	Chromosome Number (2n)	Genome Size	Genome Gene Number	R2R3-MYB Genes
Wolfberry	*L. barbarum*	24	1.67 Gb	33,581	137
Tomato	*S. lycopersicum*	24	785 Mb	34,075	133
Pepper	*C. annuum*	24	3.3 Gb	35,336	108
Potato	*S. tuberosum*	24	844 Mb	39,031	109
Eggplant	*S. melongena*	24	1.07 Gb	36,568	123

## Data Availability

The raw data of the transcriptome analysis used in this study were submitted to the Sequence Read Archive (SRA) at NCBI under Project ID PRJNA788208.

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
