# Peer review of "Genome-Wide Comparative Analysis of the R2R3-MYB Gene Family in Five Solanaceae Species and Identification of Members Regulating Carotenoid Biosynthesis in Wolfberry"

_ijms, 2022, doi:10.3390/ijms23042259_

Round 1

Reviewer 1 Report

The work of Yin et al., describe a comprehensive analysis of the R2R3-MYB family in five Solanaceae species with the aim to compare this important family of TFs among the species and highlight candidates that could possibly be involved in regulation of carotenoid biosynthesis. Integrated bioinformatic analysis and experimental verification are nicely combined and the results are of some interest to readers working with wolfberry and or carotenoids.

The rationale is easy to follow, the analyses well conducted and the discussion is nicely documented.

I just have a few minor comments that could possibly improve the manuscript.

  • Please include legends in S2, S3, S6
  • Figure 10: please include in the figure legend what does the red dots highlight.
  • Line 453--> 1000 bootstraps are mentioned in this line but in Fig S2 & S3, only up to 100 bootstrap is indicated. Please correct.
  • Line 461--> "BLSTAP" should be BLASTP

Reviewer 2 Report

Comments see in attachment fille

Reviewer 3 Report

The authors in the manuscript entitled “Genome-wide Comparative Analysis of the R2R3-MYB Gene Family in Five Solanaceae Species and Identification of Members Regulating Carotenoid Biosynthesis in Wolfberry” have identified 610 R2R3-MYB genes in five Solanaceae species along with 137 genes in wolfberry. They have performed the evolutionary relationship, gene and protein structure analysis, duplication events analysis. Further they have used RN seq data and performed the differential expression analysis and validated it by performing qRT-PCR analysis. They have shown the importance of these genes in regulating carotenoid biosynthesis in wolfberry through this study. Overall, the study has significant scientific importance and the authors have provided satisfactory evidence to support further studies. However, some comments need to be addressed which would be helpful in improving the manuscript.

  1. There are lot of typographical and grammatical error. Correct it and revise the manuscript.
  2. The authors need to add more literature in the introduction part as the gene family is quite common and much work has been done.
  3. Figure 1 can be improved as it is not clearly visible.
  4. The legends of figure 2 is not clearly explained. Explain it to make it understandable.
  5. The authors could perform the events of neo-functionalization, pseudo-functionalization etc. by performing the expression analysis of few representative duplicated gene pairs. It would be useful for studying the functional conservation.
  6. The colours demarcation is not clear in figure 4. Use bright colours.
  7. The authors are advised to increase the resolution of figure 5.
  8. The authors have not mentioned the significance of expression analysis between 12 DAP and 37 DAP stages.
  9. The authors are advised to elaborate the abbreviation used in the manuscript for making the manuscript understandable.
  10. Where is the result of carotenoid extraction? No such figure has been seen.
  11. The conclusion is less descriptive. Elaborate it.
